# ClickPrompt: CTR Models are Strong Prompt Generators for Adapting Language Models to CTR Prediction

## ABSTRACT

Click-through rate (CTR) prediction has become increasingly indispensable for various Internet applications. Traditional CTR models convert the multi-field categorical data into ID features via one-hot encoding, and extract the collaborative signals among features. Such a paradigm suffers from the problem of *semantic information loss*. Another line of research explores the potential of pretrained language models (PLMs) for CTR prediction by converting input data into textual sentences through hard prompt templates. Although semantic signals are preserved, they generally fail to capture the *collaborative information* (*e.g.*, feature interactions, pure ID features), not to mention the unacceptable *inference overhead* brought by the huge model size. In this paper, we aim to model both the semantic knowledge and collaborative knowledge for accurate CTR estimation, and meanwhile address the inference inefficiency issue. To benefit from both worlds and close their gaps, we propose a novel model-agnostic framework (*i.e.*, ClickPrompt), where we incorporate CTR models to generate interaction-aware soft prompts for PLMs. We design a prompt-augmented masked language modeling (PA-MLM) pretraining task, where PLM has to recover the masked tokens based on the language context, as well as the soft prompts generated by CTR model. The collaborative and semantic knowledge from ID and textual features would be explicitly aligned and interacted via the prompt interface. Then, we can either tune the CTR model with PLM for superior performance, or solely tune the CTR model without PLM for inference efficiency. Experiments on four real-world datasets validate the effectiveness of ClickPrompt compared with existing baselines. The source code[1] is available.

## 1 INTRODUCTION

Click-through rate (CTR) prediction serves as a key component in various online applications [7, 9, 20, 34, 58]. It aims to estimate the probability of a user's click given a specific context [35], which could be formulated as multi-field categorical data format:

$$x_i^{ID} = \underbrace{(0, ..., 1, 0)}_{Item=Jeans} \underbrace{(1, ..., 0, 0)}_{Color=Blue} ... \underbrace{(1, 0)}_{Gender=Female}. \quad (1)$$

Over the past decade, various neural CTR models have been proposed to extract collaborative knowledge and capture high-order feature interaction patterns. However, they generally suffer from the problem of **semantic information loss**. That is, the multi-field categorical data will be converted into ID features with one-hot encoding as shown in Eq. 1 (*e.g.*, "*female*" to "*01*", and "*male*" to "*10*"). Therefore, the input data of CTR models is only a collection of ID codes without any semantic information which inherently contains implicit yet beneficial correlations among features. For example, the movie "*The Avengers 4: Endgame*" is not only related to its preceding series "*The Avengers 1-3*" based on simple text similarity, but

[1]https://anonymous.4open.science/r/ClickPrompt-D6E3/

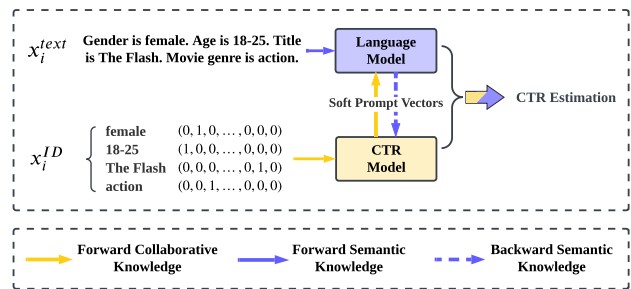

**Figure 1: The information flow of the collaborative and semantic knowledge in our proposed ClickPrompt framework. With the soft prompts serving as the bridges, the ID-based collaborative information is transmitted to the language model by forward propagation, and the text-based semantic information flows back into the CTR model via backpropagation.**

is also associated with other superhero movies (*e.g.*, "*Iron Man*" and "*Captain America*") based on latent semantic knowledge. However, once converted into one-hot ID codes, it loses the valuable semantic information, which could lead to inferior predictive performance, especially for scenarios with cold-start users/items, low-frequency long-tail features or inadequate click signals.

To this end, recent works [11, 17, 33] start to introduce pretrained language models (PLMs) to address the aforementioned semantic information loss problem. They convert multi-field categorical data into textual features with hard prompt templates instead of ID features with one-hot encoding, resulting in another textual modality for the same input sample $x_i$:

$$x_i^{text} = \text{ "User } \underline{AX529} \text{ is a } \underline{female}. \underline{Her} \text{ profession is a } \underline{nurse}. \\ \underline{Her} \text{ location is } \underline{New York}. \underline{She} \text{ is recommended item } \underline{CF173}, \quad (A) \\ \text{a pair of } \underline{jeans} \text{ in color } \underline{blue}. \text{ ''}$$

In Template A shown above, the underlined words or phrases need to be dynamically filled in according to the input sample $x_i$. In this way, these works preserve the semantic information by formulating CTR prediction as either a sequence-to-sequence task [6, 10, 62] or a binary classification task [23, 38]. PLMs possess a vast volume of open-world knowledge from the pretraining corpora and even show impressive emergent abilities (*e.g.*, logical reasoning) if the parameter size scales up, which helps to capture the semantic information. Nevertheless, simply adapting PLMs for CTR estimation generally suffers from two limitations, *i.e.*, **predictive inaccuracy** and **inference inefficiency**.

The predictive inaccuracy is mainly caused by the disability of PLMs in modeling collaborative knowledge [33, 57]. *First*, there exists a kind of pure ID features that inherently contains no semantic information (*e.g.*, item ID, user ID). The tokenization results of these pure ID features are actually meaningless to PLMs (*e.g.*, user ID AX529 might be tokenized as [AX, 52, 9]). *Second*, PLMs struggle to explicitly capture feature interactions, since all the field-wise

features are assembled linearly as textual sentences via templates and then broken down into word tokens [27]. PLMs can model the contextual semantic information among word tokens, but it loses the field-level views of feature interactions which are essential for CTR prediction. Preliminary works address such challenges by introducing additional embedding tables [10], maintaining a set of adapter modules [11], and seeking for better ID indexing strategies [18]. However, the collaborative knowledge embedded among ID features is still under-exploited, *e.g.*, the field-aware feature interactions are not explicitly maintained.

The inference inefficiency issue stems from the intrinsic characteristics of pretrained language models, where bigger model size is required for better language understanding ability [33, 61]. Adapting PLM will greatly increase the computational cost and inference time due to its large-scale stacked attention-based transformer layers. This is unacceptable for real-world time-sensitive online services, where a request should be responded within tens of milliseconds. Many works [16, 46, 59] tend to adopt the whole PLM for training, and pre-cache the output representations of PLM for inference acceleration, which heavily requires storage and computational resources, as well as engineering efforts. Moreover, the pre-caching operation might impair the real-time property of recommender systems and thus hurt the predictive performance.

In this paper, we aim to capture both the semantic knowledge and collaborative knowledge for accurate CTR prediction, while tackling the inference inefficiency problem in the meantime. To this end, we propose a novel framework named **ClickPrompt**, where we regard CTR models[2] as soft prompt generators for PLMs. Specifically, we maintain a CTR model and a pretrained language model, which take as inputs the ID features $x_i^{ID}$ and textual features $x_i^{text}$, respectively. A prompt generation layer is placed upon the CTR model to produce learnable soft prompt vectors, which will be fed as prefix states into each layer of PLM. ClickPrompt follows the pretrain-finetune learning schemes [8, 35]. We design a *prompt-augmented masked language modeling* (PA-MLM) pretraining task. To be specific, we first adopt the token-masking strategy from BERT [8] to obtain the masked textual features $\hat{x}_i^{text}$. Then, PLM is required to recover the corrupted textual features $\hat{x}_i^{text}$ based on the text context, as well as the soft prompts generated from ID features $x_i^{ID}$. As shown in Figure 1, with the soft prompts as the bridges, the ID-based collaborative knowledge will be passed to PLM through forward propagation, and the text-based semantic knowledge would flow back into the CTR model via backpropagation. After pretraining, we propose two different finetuning strategies for CTR prediction:

- **Finetune with PLM**. We could tune the CTR model and PLM as a whole, where they are connected by the prompt generation layer. The collaborative knowledge from CTR model and semantic knowledge from PLM would explicitly align and interact with each other through the soft prompt interface, resulting in superior CTR performance.
- **Finetune w/o PLM**. To further tackle the inference inefficiency issue, we can solely finetune the CTR model without PLM. PA-MLM pretraining has provided semantic-aware parameter initialization for downstream CTR finetuning, which promotes the

---

[2]In this paper, unless specified otherwise, the phrase "CTR model" is referred to as traditional CTR models that take one-hot ID features as inputs.

final performance without altering the CTR model structure or adding extra inference costs.

ClickPrompt serves as a model-agnostic framework that is compatible with various CTR models and pretrained language models. The main contributions of this paper are concluded as follows:

- We propose a novel framework (*i.e.*, ClickPrompt), where CTR models serve as the soft prompt generators for PLMs. A prompt-augmented masked language modeling pretraining (PA-MLM) task is designed to model the mutual interaction and explicit alignment between the collaborative and semantic knowledge via the soft prompt interface, which significantly improves the downstream CTR performance.
- ClickPrompt is model-agnostic and compatible with various CTR models and PLMs. Moreover, by solely finetuning the CTR model, ClickPrompt can enhance the predictive accuracy without altering the CTR model structure or adding extra inference costs.
- Extensive experiments on four real-world public datasets demonstrate the superiority of our proposed ClickPrompt, compared with existing baseline models.

## 2 PRELIMINARIES

### 2.1 Traditional CTR Prediction

Without loss of generality, the basic form of CTR prediction casts a binary classification problem over multi-field categorical data. Each data sample contains $F$ fields with each field taking one single value from multiple categories, and can be represented by $(x_i, y_i)$. In traditional CTR prediction, we apply one-hot encoding to convert $x_i$ into a sparse vector $x_i^{ID}$ as shown in Eq. 1, and maintain $y_i \in \{1, 0\}$ as the ground-true label (click or not).

CTR models estimate the click probability $P(y_i = 1|x_i)$ for each instance. According to [35, 54, 63], the structure of most traditional neural CTR models can be abstracted into three layers: (1) embedding layer, (2) feature interaction layer, and (3) prediction layer.

**Embedding layer** transforms the sparse one-hot input $x_i^{ID}$ into low-dimensional dense embedding vectors $\mathbf{E}_i = [v_{i1}; v_{i2}; \dots; v_{iF}] \in \mathbb{R}^{F \times d}$, where $d$ is the embedding size, and $F$ is the number of fields.

**Feature interaction layer**, as the key functional module of CTR models, is intended to capture the second- or higher-order feature interactions with various operations (*e.g.*, attention, product). This layer would generate a compact representation $q_i$ based on the dense embedding vectors $\mathbf{E}_i$ for the data instance $x_i$.

**Prediction layer** calculates the click probability $\hat{y}_i = P(y_i = 1|x_i)$ based on the representation $q_i$ produced by the feature interaction layer. It is usually a linear layer or an MLP module followed by a sigmoid function $\sigma(x) = 1/(1 + e^{-x})$.

After the prediction layer, the CTR model is trained in an end-to-end manner with the binary cross-entropy (BCE) loss:

$$\mathcal{L} = -\frac{1}{N} \sum_{i=1}^{N} \left[ y_i \log \hat{y}_i + (1 - y_i) \log(1 - \hat{y}_i) \right], \quad (2)$$

where $N$ is the number of training samples.

### 2.2 PLM-based CTR Prediction

With the rising of pretrained language models (PLMs), researchers exploit the semantic-related abilities of PLMs to solve the CTR estimation problem. Different from traditional CTR prediction, the

input $x_i$ is transformed into textual sentences $x_i^{text}$ via hard prompt templates as illustrated in Template A. According to the task genre and ground-truth label formulation, PLM-based CTR prediction can be roughly divided into two categories [33, 57].

The first one [1, 38, 46] regards CTR prediction as a binary text classification task, where the ground-truth labels are still the same as traditional settings (*i.e.*, $y_i \in \{0, 1\}$). They leverage PLMs to extract the dense representation $q_i^{text}$ of textual input $x_i^{text}$, which is followed by the prediction layer for click estimation. BCE loss is adopted for optimization.

$$\hat{y}_i = \sigma \left( \text{MLP} \left( \text{PLM} \left( x_i^{text} \right) \right) \right) \in (0, 1). \tag{3}$$

The second category [6, 10, 17] views CTR prediction as a sequence-to-sequence task, where the ground-truth labels are transformed into binary key words (*e.g.*, yes/no, good/bad). They utilize encoder-decoder or decoder-only PLMs to follow instructions and answer a binary question (*e.g.*, *Will the user favor the item?*) appended behind the textual input $x_i^{text}$. PLMs could be either frozen for zero-shot settings, or finetuned via causal language modeling.

In this paper, we generally focus on the first category. That is, we place an MLP module upon the textual representation $q_i^{text}$ produced by the PLM.

## 3 METHODOLOGY

In this section, we introduce the details of model architecture and learning strategy of our proposed ClickPrompt framework.

### 3.1 Overview of ClickPrompt

As depicted in Figure 2, the model architecture design of Click-Prompt can be mainly divided into three stages: (1) modality transformation, (2) prompt generation, and (3) prompt fusion.

*Firstly*, the modality transformation layer converts the input data $x_i$ into one-hot ID features $x_i^{ID}$ and textual features $x_i^{text}$, respectively. *Secondly*, ID features $x_i^{ID}$ are fed into the CTR model followed by a prompt generation layer to produce independent soft prompt vectors. *Finally*, during the prompt fusion stage, the soft prompts serve as prefix hidden states at each transformer layer of PLM, which allows for the explicit alignment between collaborative and semantic knowledge.

As for the learning strategy, ClickPrompt adopts the common pretrain-finetune scheme. We first design a prompt-augmented masked language modeling (PA-MLM) task for pretraining, where PLM is required to recover the masked tokens based on text contexts as well as soft prompts generated by the CTR model. After pretraining, we can conduct supervised finetuning either with or without PLM. The former enables explicit interaction between collaborative and semantic information for superior performance, while the latter addresses the inference inefficiency issue.

Hereinafter, we omit the detailed structures of the CTR model and PLM, since ClickPrompt acts as a model-agnostic framework for both of them.

### 3.2 Modality Transformation

The modality transformation layer converts the input $x_i$ into two different modalities (*i.e.*, ID features $x_i^{ID}$ and textual features $x_i^{text}$

respectively). The ID features $x_i^{ID}$ are obtained via one-hot encoding as shown in Eq. 1. As for the textual features $x_i^{text}$, previous works [3, 15] suggest that sophisticated templates for tabular data might mislead the model and make it fail to grasp the key information among the texts. Therefore, we employ the following simple "*what is what*" hard prompt template:

$$x_{i,j}^{text} = \left[ f_j^{name}, \text{"is"}, f_{i,j}, \text{"."} \right], \ j = 1, ..., F,$$
$$x_i^{text} = \left[ x_{i,1}^{text}, x_{i,2}^{text}, \cdots, x_{i,F}^{text} \right], \tag{4}$$

where $f_j^{name}$ is the field name of $j$-th field, $f_{i,j}$ is the feature value of the $j$-th field in $i$-th data instance, and $[\cdot]$ denotes the conjunct operator to concatenate elements in the list with white spaces " ".

### 3.3 Prompt Generation

The prompt generation stage aims to encode the ID features $x_i^{ID}$ into independent soft prompt vectors that contain rich collaborative knowledge for later fusion. As described in Section 2.1, we feed the ID input $x_i^{ID}$ through the embedding and feature interaction (FI) layer of the CTR model to obtain the compact representation $q_i$:

$$q_i = \text{FI\_Layer}(\text{Embed\_Layer}(x_i^{ID})). \tag{5}$$

Then, we maintain a set of parallel projection networks $\{g_{l,k}(\cdot)\}$ for soft prompt generation:

$$p_{i,l,k} = g_{l,k}(q_i), \ 1 \le l \le L, \ 1 \le k \le K, \tag{6}$$

where $p_{i,l,k}$ denotes the $k$-th prompt vector at the $l$-th layer of PLM. $L$ is the number of transformer layer of PLM, and $K$ is the number of soft prompts per layer. Each projection network $g_{l,k}(\cdot)$ is designed as a multi-layer perceptron (MLP), facilitating dimensionality consistency and space transformation.

### 3.4 Prompt Fusion

As shown in Figure 2, the obtained soft prompts would serve as prefix hidden states at each transformer layer of PLM. To be specific, the textual features $x_i^{text}$ are tokenized into $Z$ word tokens, and the $l$-th layer of PLM can be formulated as:

$$[h_{i,l+1,z}]_{z=1}^{Z} = \text{Transformer}_l \left( [p_{i,l,k}]_{k=1}^{K} \oplus [h_{i,l,z}]_{z=1}^{Z} \right), \tag{7}$$

where $[h_{i,l,z}]_{z=1}^{Z}$ are the hidden states of tokens at each layer $l$. In this way, through the self-attention mechanism of each transformer layer, the collaborative signals from the CTR model can be explicitly aligned and fused with the semantic knowledge from the text side via the prompt interface.

Finally, after the $L$ layer propagation, we apply the pooling & prediction layer upon the output states of PLM:

$$\text{MLP} \left( \text{Pooling} \left( [h_{i,L+1,z}]_{z=1}^{Z} \right) \right). \tag{8}$$

The output dimensionality, as well as the following activation and loss function, depends on the task and learning strategy we adopt, which will be further discussed in Section 3.5.

### 3.5 Learning Strategy

As shown in Figure 2, ClickPrompt employs the common pretrain-finetune scheme for learning strategy. Specifically, we first propose prompt-augmented masked language modeling (PA-MLM) as

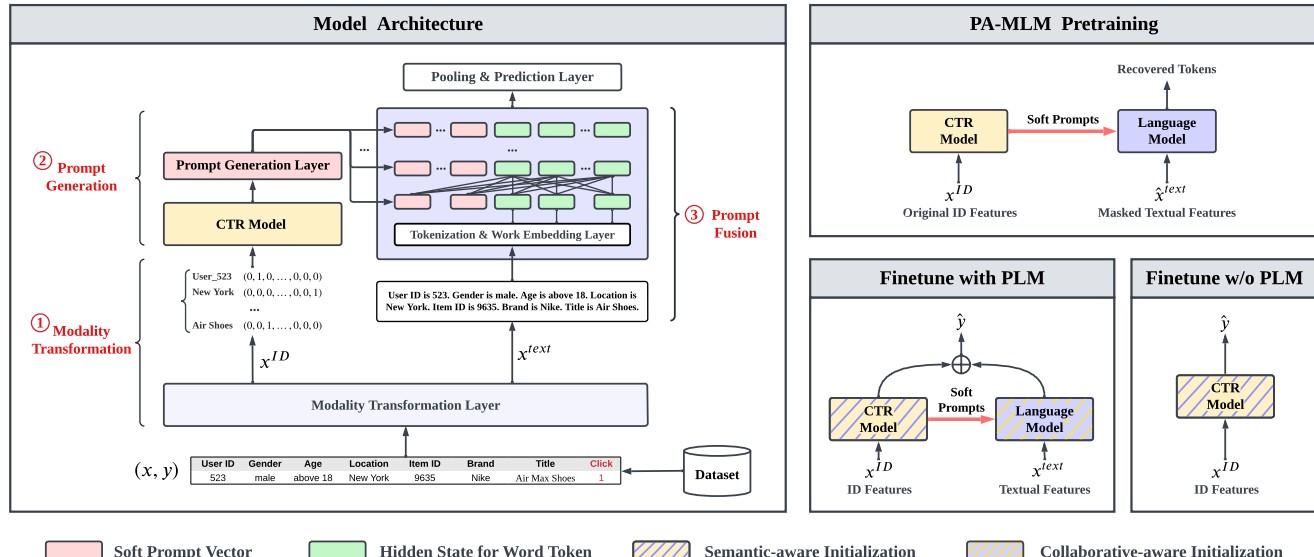

**Figure 2: The illustration of the model architecture and learning strategy for our proposed ClickPrompt framework.**

the pretraining task to intermingle the collaborative and semantic knowledge via the linkage of soft prompts, resulting in improved parameter initialization. Next, we could either perform supervised finetuning *with PLM* for superior CTR performance, or solely finetune the CTR model *without PLM* to preserve both the improved predictive accuracy and inference efficiency.

*3.5.1 Prompt-augmented Masked Language Modeling.* As shown in Figure 2, we propose to apply token masking on the textual features $x_i^{text}$ to obtain corrupted textual inputs $\hat{x}_i^{text}$, while preserving the original ID features $x_i^{ID}$. Then, PLM is required to recover the masked tokens based on the language context, together with the soft prompts generated from the intact ID features. Therefore, the pooling & prediction layer in Eq. 8 is designed as the classical decoder module of language models, which is followed by a softmax function and cross-entropy loss. Following [8, 43], we uniformly sample 15% tokens for each input $x_i^{text}$, and perform three different operations with ratio 8:1:1, *i.e.*, (1) [MASK] replacement, (2) random word replacement, and (3) keeping unchanged.

To complete such a cloze task over masked tokens, PLM has to extract and incorporate the corresponding "right answer" embedded in the soft prompts, resulting in fine-grained alignments between the CTR model and PLM towards the same input $x_i$.

*3.5.2 Finetuning with PLM.* Obviously, we can retain the whole model structure, and continue supervised finetuning for downstream CTR prediction task. As illustrated in Figure 2, we integrate the predictions from both CTR model and PLM, while they are explicitly interacted with soft prompt vectors:

$$\hat{y}_i^{CTR} = \text{MLP}(q_i) \in \mathbb{R},$$
$$\hat{y}_i^{PLM} = \text{MLP}\left(\text{Pooling}\left([h_{i,L+1,z}]_{z=1}^Z\right)\right) \in \mathbb{R}, \quad (9)$$
$$\hat{y}_i = \sigma\left(\alpha \times \hat{y}_i^{CTR} + (1-\alpha) \times \hat{y}_i^{PLM}\right),$$

where $\alpha$ is a learnable parameter to balance the weights of predictions, and $\sigma(\cdot)$ is the sigmoid function. In this way, the collaborative

and semantic knowledge from two modalities are thoroughly connected and intertwined during finetuning, thus leading to superior CTR performance.

*3.5.3 Finetuning without PLM.* To further address the inference inefficiency issue, as depicted in Figure 2, we could solely finetune the CTR model without PLM. We have injected the semantic knowledge from PLM into the CTR model through backpropagation during PA-MLM pretraining. Hence, such a semantic-aware parameter initialization would enable implicit interactions between collaborative and semantic knowledge, enhancing CTR performance without altering the CTR model structure or adding extra inference cost:

$$\hat{y}_i = \sigma(\text{MLP}(q_i)) \in \mathbb{R}. \quad (10)$$

For both two finetuning strategies, we apply the binary cross entropy loss over the estimated click probability as shown in Eq. 2.

## 4 EXPERIMENT

In this section, we conduct extensive experiments to answer the following research questions:

**RQ1** How does ClickPrompt perform compared with existing baseline models?

**RQ2** Is ClickPrompt compatible with various CTR models and pretrained language models?

**RQ3** What are the influences of different model configurations for ClickPrompt?

**RQ4** How does ClickPrompt perform in scenarios with long-tail low-frequency users or items?

**Table 1: The dataset statistics**

| Dataset | #Training | #Validation | #Test | #Fields | #Features |
|---|---|---|---|---|---|
| Movielens-1M | 591,208 | 73,902 | 73,902 | 8 | 17,251 |
| BookCrossing | 824,936 | 103,117 | 103,118 | 8 | 722,234 |
| Amazon-Toys | 1489782 | 186,223 | 186,223 | 5 | 371,813 |
| GoodReads | 16,097,632 | 2,012,204 | 2,012,204 | 15 | 4,565,430 |

**Table 2: The overall performance comparison of different models. The best result is given in bold, while the second-best value is underlined. We also use wavy underline to denote the best baseline performance. _Rel.Impr_ denotes the relative AUC improvement rate of our proposed ClickPrompt$_{with\ PLM}$ against each baseline model. The symbol * indicates statistically significant improvement of ClickPrompt over the best baseline model with $p < 0.001$.**

| Model | Movielens-1M | | | BookCrossing | | | Amazon-Toys | | | GoodReads | | |
|---|---|---|---|---|---|---|---|---|---|---|---|---|
| | AUC | Log Loss | Rel.Impr | AUC | Log Loss | Rel.Impr | AUC | Log Loss | Rel.Impr | AUC | Log Loss | Rel.Impr |
| FM | 0.8371 | 0.4090 | 1.53% | 0.7871 | 0.5202 | 2.02% | 0.6668 | 0.4059 | 1.30% | 0.7614 | 0.5190 | 1.85% |
| DNN | 0.8413 | 0.3944 | 1.02% | 0.7940 | 0.5124 | 1.13% | 0.6686 | 0.3982 | 1.03% | 0.7685 | 0.5082 | 0.91% |
| DeepFM | 0.8443 | 0.3915 | 0.66% | 0.7959 | 0.5106 | 0.89% | 0.6692 | 0.3978 | 0.94% | 0.7690 | 0.5136 | 0.85% |
| xDeepFM | 0.8435 | 0.3950 | 0.76% | 0.7943 | 0.5122 | 1.10% | 0.6681 | 0.3967 | 1.11% | 0.7697 | 0.5072 | 0.75% |
| IPNN | 0.8437 | 0.3926 | 0.73% | 0.7953 | 0.5111 | 0.97% | 0.6687 | 0.3980 | 1.02% | 0.7722 | 0.5148 | 0.43% |
| DCN | 0.8423 | 0.3964 | 0.90% | 0.7952 | 0.5116 | 0.98% | 0.6688 | 0.3964 | 1.00% | 0.7693 | 0.5074 | 0.81% |
| AutoInt | 0.8399 | 0.4004 | 1.19% | 0.7954 | 0.5113 | 0.96% | 0.6678 | 0.3977 | 1.15% | 0.7682 | 0.5084 | 0.95% |
| FiGNN | 0.8399 | 0.3991 | 1.19% | 0.7970 | 0.5105 | 0.75% | 0.6700 | 0.3947 | 0.82% | 0.7667 | 0.5094 | 1.15% |
| FGCNN | 0.8416 | 0.3957 | 0.99% | 0.7985 | 0.5082 | 0.56% | 0.6675 | 0.3978 | 1.20% | 0.7705 | 0.5064 | 0.65% |
| DCNv2 | 0.8439 | 0.3954 | 0.71% | 0.7970 | 0.5096 | 0.75% | 0.6701 | 0.3961 | 0.81% | 0.7711 | 0.5059 | 0.57% |
| CTR-BERT | 0.8296 | 0.4208 | 2.45% | 0.7848 | 0.5268 | 2.32% | 0.6649 | 0.3988 | 1.59% | 0.7457 | 0.5292 | 4.00% |
| P5 | 0.8173 | 0.4171 | 3.99% | 0.7695 | 0.5360 | 4.35% | 0.6470 | 0.4018 | 4.40% | 0.7367 | 0.5531 | 5.27% |
| PTab | 0.8353 | 0.4081 | 1.75% | 0.7979 | 0.5208 | 0.64% | 0.6685 | 0.3995 | 1.05% | 0.7566 | 0.5203 | 2.50% |
| CTRL | 0.8453 | 0.3932 | 0.54% | 0.7992 | 0.5092 | 0.48% | 0.6704 | 0.3960 | 0.76% | 0.7735 | 0.5038 | 0.26% |
| ClickPrompt$_{w/o\ PLM}$ | 0.8467* | 0.3939 | - | 0.8013* | 0.5051* | - | 0.6719* | 0.3933* | - | 0.7744* | 0.5030* | - |
| ClickPrompt$_{with\ PLM}$ | **0.8499*** | **0.3905*** | - | **0.8030*** | **0.5037*** | - | **0.6755*** | **0.3890*** | - | **0.7755*** | **0.5022*** | - |

## 4.1 Experiment Setups

*4.1.1 Datasets.* Since PLM-based CTR prediction requires datasets to maintain the original semantic/textual features, instead of anonymous feature IDs, we select four real-world public datasets from different recommendation scenarios (*i.e.*, MovieLens-1M[3], BookCrossing[4], Amazon-Toys[5], and GoodReads[6]). All datasets are divided into training, validation, and testing sets with proportion 8:1:1 according to the global timestamps. The basic statistics of these four datasets are summarized in Table 1. More detailed information about the datasets and data preprocessing is given in Appendix A

*4.1.2 Evaluation Metrics.* To evaluate the performance of CTR prediction methods, we adopt AUC (Area under the ROC curve) and Log Loss (binary cross-entropy loss) as the evaluation metrics. Slightly higher AUC or lower Log Loss (*e.g.*, 0.001) can be regarded as significant improvement in CTR prediction [32, 54, 56]

*4.1.3 Baselines.* For traditional CTR models, we select baselines with different feature interaction operators, including FM [52], DNN, DeepFM [12], xDeepFM [32], PNN [48], DCN [55], AutoInt [53], FiGNN [30], FGCNN[37], and DCNv2 [56]. For PLM-based CTR models, we choose CTR-BERT [46], P5 [10], PTab [38], CTRL [27] as the representative baselines.

*4.1.4 Implementation Details.* We adopt AdamW as the optimizer. For prompt-augmented masked language modeling pretraining, we set batch size to 1024 and learning rate to $5 \times 10^{-5}$. The warm-up ratio is selected from $\{0, 0.05, 0.1\}$. The number of pretraining epoch is 20. For the finetuning phase, the batch size is set to 256 for Movielens-1M, 256 for BookCrossing, 1024 for AZ-Toys, and 4096 for GoodReads. The learning rate for CTR model part

[3]https://grouplens.org/datasets/movielens/1m/
[4]http://www2.informatik.uni-freiburg.de/~cziegler/BX/
[5]https://cseweb.ucsd.edu/~jmcauley/datasets.html
[6]https://mengtingwan.github.io/data/goodreads.html

is $1 \times 10^{-3}$, while the learning rate for PLM part is selected from $\{0, 3 \times 10^{-5}, 5 \times 10^{-5}\}$. Setting the learning rate for PLM as zero means that we freeze the language model and only update the CTR model. The projection network $g_{l,k}$ for prompt generation is a tanh-activated two-layer MLP with hidden size equal to the embedding size of PLM. The number of prompts per layer $K$ is selected from $\{1, 3, 5, 7\}$. Since ClickPrompt is a model-agnostic framework, we choose DCNv2 [56] as the CTR model and RoBERTa-base [43] as the pretrained language model, unless otherwise specified. Finally, we adopt the model at the iteration with the highest validation AUC for evaluation in the testing set. We also provide the detailed hyperparameter settings for each baseline models in Appendix B

## 4.2 Overall Performance (RQ1)

We compare the overall performance of our proposed ClickPrompt with the selected baseline models. Note that we choose DCNv2 as the CTR model and RoBERTa-base as the pretrained language model. The results are reported in Table 2, from which we can obtain the following observations:

- Traditional CTR models show significantly better performance over PLM-based CTR models, except for CTRL. This indicates that the collaborative information embeded among feature crossing patterns is crucial for CTR prediction, and solely relying on semantic knowledge from textual inputs might lead to inferior performance, which is consistent with the results in [27].
- CTRL generally achieves the best performance among all the baseline models. CTRL adopts the CLIP-based framework [50], and distills the semantic knowledge from PLM into the CTR model via contrastive pretraining. However, the contrastive objective could only provide coarse-grained instance-level supervisions for implicit alignment and late interaction upon the final representations of PLM and CTR model, resulting in relatively inferior performance compared with our proposed ClickPrompt.

**Table 3: The model compatibility analysis of our proposed ClickPrompt over different CTR models and PLMs. *N/A* means to train the raw CTR model from scratch without ClickPrompt. For each CTR model, we denote the best result in bold, and underline the second-best value. *Rel.Impr* denotes the relative AUC improvement rate against the raw CTR model (*i.e., N/A*). The improvements are statistically significant with $p < 0.001$ against the corresponding raw CTR models (*i.e., N/A*).**

| CTR Model | Finetuning | Language Model | Movielens-1M | | | BookCrossing | | | Amazon-Toys | | |
|---|---|---|---|---|---|---|---|---|---|---|---|
| | | | AUC | Log Loss | Rel.Impr | AUC | Log Loss | Rel.Impr | AUC | Log Loss | Rel.Impr |
| DCNv2 | | N/A | 0.8439 | 0.3954 | - | 0.7970 | 0.5096 | - | 0.6701 | 0.3961 | - |
| | w/o PLM | TinyBERT | 0.8464 | 0.3943 | 0.30% | 0.7997 | 0.5070 | 0.34% | 0.6705 | 0.3956 | 0.06% |
| | | RoBERTa-base | 0.8467 | 0.3939 | 0.33% | 0.8013 | 0.5051 | 0.54% | 0.6719 | **0.3933** | 0.27% |
| | | RoBERTa-large | **0.8476** | **0.3920** | 0.44% | **0.8017** | **0.5047** | 0.59% | **0.6723** | 0.3939 | 0.33% |
| | with PLM | TinyBERT | 0.8470 | 0.3933 | 0.37% | 0.8003 | 0.5063 | 0.41% | 0.6732 | 0.3943 | 0.46% |
| | | RoBERTa-base | **0.8499** | **0.3905** | 0.71% | 0.8030 | 0.5037 | 0.75% | 0.6755 | **0.3890** | 0.81% |
| | | RoBERTa-large | 0.8498 | 0.3918 | 0.70% | **0.8032** | **0.5034** | 0.78% | **0.6759** | 0.3893 | 0.87% |
| AutoInt | | N/A | 0.8399 | 0.4004 | - | 0.7954 | 0.5113 | - | 0.6678 | 0.3977 | - |
| | w/o PLM | TinyBERT | 0.8422 | 0.3995 | 0.27% | 0.7967 | 0.5098 | 0.16% | 0.6714 | 0.3948 | 0.54% |
| | | RoBERTa-base | 0.8439 | 0.3967 | 0.48% | 0.7981 | 0.5091 | 0.34% | 0.6724 | 0.3944 | 0.69% |
| | | RoBERTa-large | **0.8454** | **0.3965** | 0.65% | **0.7989** | **0.5084** | 0.44% | **0.6732** | **0.3918** | 0.81% |
| | with PLM | TinyBERT | 0.8458 | 0.3915 | 0.70% | 0.7981 | 0.5081 | 0.34% | 0.6728 | 0.3943 | 0.75% |
| | | RoBERTa-base | 0.8465 | 0.3912 | 0.79% | 0.8004 | 0.5076 | 0.63% | 0.6760 | 0.3924 | 1.23% |
| | | RoBERTa-large | **0.8481** | **0.3893** | 0.98% | **0.8009** | **0.5070** | 0.69% | **0.6767** | **0.3893** | 1.33% |
| DNN | | N/A | 0.8413 | 0.3944 | - | 0.7940 | 0.5124 | - | 0.6686 | 0.3982 | - |
| | w/o PLM | TinyBERT | 0.8435 | 0.3944 | 0.26% | 0.7960 | 0.5114 | 0.25% | 0.6700 | 0.3956 | 0.21% |
| | | RoBERTa-base | 0.8448 | 0.3929 | 0.42% | 0.7972 | 0.5097 | 0.40% | 0.6704 | 0.3943 | 0.27% |
| | | RoBERTa-large | **0.8455** | 0.3927 | 0.50% | **0.7985** | **0.5081** | 0.57% | **0.6710** | 0.3942 | 0.36% |
| | with PLM | TinyBERT | 0.8446 | 0.3925 | 0.39% | 0.7971 | 0.5093 | 0.39% | 0.6732 | 0.3946 | 0.69% |
| | | RoBERTa-base | 0.8455 | **0.3909** | 0.50% | 0.7994 | 0.5080 | 0.68% | 0.6742 | 0.3935 | 0.84% |
| | | RoBERTa-large | **0.8462** | 0.3914 | 0.58% | **0.7999** | **0.5070** | 0.74% | **0.6745** | 0.3930 | 0.88% |

- ClickPrompt$_{with\ PLM}$ achieves significant improvements over all the baseline models, which validates the effectiveness of explicit alignment and early interaction between collaborative and semantic knowledge through the soft prompt interface.
- ClickPrompt$_{w/o\ PLM}$ generally wins the second place, significantly outperforming baseline methods without altering the model structure of DCNv2. This demonstrates the importance of semantic-aware parameter initialization brought by PA-MLM pretraining. By sacrificing the opportunity for explicit interaction with semantic signals during downstream finetuning, ClickPrompt$_{w/o\ PLM}$ successfully promotes the predictive accuracy without increasing the inference latency.

- As the model size of PLM continues to grow, the performance improvement over the raw CTR model brought by ClickPrompt also gradually increases, except for few cases. Larger pretrained language models possess a broader range of open-world knowledge, which can benefit the fusion and alignment between semantic and collaborative signals.
- Although we observe the phenomenon that performance continues to increase with the language model size, a larger volume of PLM does not necessarily lead to a *proportional improvement* in CTR predictive performance. Therefore, considering the training overhead, we suggest RoBERTa-base to be a more proper and economic choice for ClickPrompt to balance the performance gain and training cost when involving PLMs.

## 4.3 Model Compatibility (RQ2)

To investigate the model compatibility, we apply the ClickPrompt framework over different backbones in terms of both CTR models and PLMs. For CTR models, we select DCNv2 [56], AutoInt [53] and DNN, which represent different type of feature interaction operators. For PLMs, we choose the following three backbones with different model sizes: TinyBERT (14.5M) [21], RoBERTa-base(125M) [43], and RoBERTa-large(335M) [43]. We conduct the model compatibility experiments on Movielens-1M, BookCrossing, and Amazon-Toys datasets. The results are reported in Table 3, from which we can obtain the following observations:

- ClickPrompt is able to achieve significant improvement over the raw CTR model (*i.e.*, N/A) for all the backbones, which demonstrates its superior model compatibility in terms of both CTR models and PLMs.

## 4.4 Ablation Study (RQ3)

We analyze the impact of hyperparameters and different configurations in ClickPrompt, including the prompt strategy, and the collaborative & semantic knowledge fusion strategy. In this section, we select DCNv2, AutoInt and DNN as the backbone CTR models, and choose RoBERTa-base as the PLM backbone. Experiments are conducted on Movielens-1M, BookCrossing, and Amazon-Toys datasets under the *finetuning-with-PLM* strategy.

*4.4.1 Prompt Strategy.* We compare two different prompt strategies shown in Figure 3, and report the results in Table 4. We observe that the layerwise prompt strategy consistently outperforms that without layerwise prompting. If the prompt vectors are only placed at the shallow input layer, the collaborative knowledge from CTR model might be overwhelmed during the PLM forwarding, thus

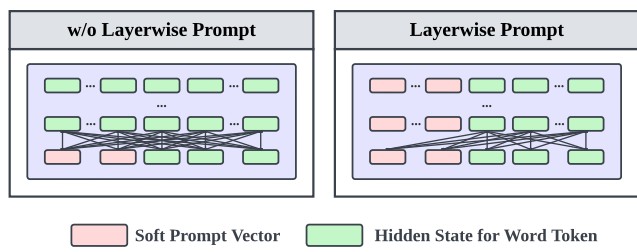

**Figure 3: The illustration of prompt strategy variants. We can either only insert the prompts at the first input layer (left), or perform layerwise soft prompts for PLM (right). Note that ClickPrompt adopts the latter prompt strategy.**

**Table 4: The ablation study of the prompt strategies illustrated in Figure 3. The best results are given in bold. *Rel.Impr* denotes the relative AUC improvement rate of *Layerwise* strategy against *w/o-Layerwise* strategy.**

| Dataset | CTR Model | Prompt Strategy | | | | Rel.Impr |
| | | w/o Layerwise | | Layerwise | | |
| | | AUC | Log Loss | AUC | Log Loss | |
|---|---|---|---|---|---|---|
| Movielens-1M | DCNv2 | 0.8468 | 0.3948 | **0.8499** | **0.3905** | 0.37% |
| | AutoInt | 0.8445 | 0.3946 | **0.8465** | **0.3912** | 0.24% |
| | DNN | 0.8433 | 0.3959 | **0.8455** | **0.3909** | 0.26% |
| BookCrossing | DCNv2 | 0.7993 | 0.5075 | **0.8030** | **0.5037** | 0.46% |
| | AutoInt | 0.7982 | 0.5091 | **0.8004** | **0.5076** | 0.28% |
| | DNN | 0.7981 | 0.5109 | **0.7994** | **0.5080** | 0.16% |
| Amazon-Toys | DCNv2 | 0.6712 | 0.3945 | **0.6755** | **0.3890** | 0.64% |
| | AutoInt | 0.6702 | 0.4006 | **0.6760** | **0.3924** | 0.87% |
| | DNN | 0.6695 | 0.3962 | **0.6742** | **0.3935** | 0.70% |

**Table 5: The ablation study on the fusion strategy for collaborative and semantic knowledge. The best values are given in bold, while the second best values are underlined.**

| CTR Model | Variant | Movielens-1M | | BookCrossing | | Amazon-Toys | |
| | | AUC | Log Loss | AUC | Log Loss | AUC | Log Loss |
|---|---|---|---|---|---|---|---|
| DCNv2 | ClickPrompt | **0.8499** | **0.3905** | **0.8030** | **0.5037** | **0.6755** | **0.3890** |
| | w/o Prompt | 0.8470 | 0.3939 | 0.8016 | 0.5049 | 0.6735 | 0.3922 |
| | w/o Pretrain | 0.8439 | 0.3949 | 0.8008 | 0.5057 | 0.6727 | 0.3917 |
| | w/o Both | 0.8438 | 0.3960 | 0.7993 | 0.5073 | 0.6706 | 0.3966 |
| AutoInt | ClickPrompt | **0.8465** | **0.3912** | **0.8004** | **0.5076** | **0.6760** | **0.3924** |
| | w/o Prompt | 0.8443 | 0.3992 | 0.7999 | 0.5082 | 0.6722 | 0.3985 |
| | w/o Pretrain | 0.8450 | 0.3953 | 0.7987 | 0.5092 | 0.6720 | 0.3945 |
| | w/o Both | 0.8448 | 0.3967 | 0.7982 | 0.5127 | 0.6699 | 0.3978 |
| DNN | ClickPrompt | **0.8455** | **0.3909** | **0.7994** | **0.5080** | **0.6742** | **0.3935** |
| | w/o Prompt | 0.8437 | 0.3959 | 0.7988 | 0.5079 | 0.6699 | 0.3979 |
| | w/o Pretrain | 0.8445 | 0.3951 | 0.7972 | 0.5123 | 0.6718 | 0.3947 |
| | w/o Both | 0.8441 | 0.3953 | 0.7973 | 0.5128 | 0.6698 | 0.3996 |

leading to unbalanced interactions with semantic knowledge and consequently inferior performance.

*4.4.2 Collaborative & Semantic Knowledge Fusion Strategy.* In Click-Prompt, there are two key technical points for the interaction and alignment between the collaborative and semantic knowledge.

(1) From the *model architecture* perspective, the layerwise soft prompts serve as the bridge for explicit interactions between CTR models and PLMs.

**Table 6: The performance of DCNv2 and ClickPrompt (DCNv2 as backbone) for the long-tail user/item problems on MovieLens-1M dataset. The best results are given in bold. *Rel.Impr* denotes the relative AUC improvement rate.**

| Long-tail User ? | Long-tail Item ? | DCNv2 | | ClickPrompt | | Rel.Impr |
| | | AUC | Log Loss | AUC | Log Loss | |
|---|---|---|---|---|---|---|
| ✔ | ✔ | 0.6000 | 0.6624 | **0.6500** | **0.6038** | 8.33% |
| ✘ | ✔ | 0.6886 | 0.6930 | **0.7003** | **0.6888** | 1.70% |
| ✔ | ✘ | 0.8149 | 0.3977 | **0.8186** | **0.3916** | 0.45% |
| ✘ | ✘ | 0.8485 | 0.3978 | **0.8520** | **0.3926** | 0.41% |

(2) From the *learning strategy* perspective, PA-MLM pretraining task forces PLM to extract and incorporate the useful collaborative information embedded in prompt vectors, resulting in fine-grained alignments.

Hence, we compare ClickPrompt with the following three variants:

- **w/o Prompt**. We retain the PA-MLM pretraining phase, but remove the prompt interface between the CTR model and PLM during the finetuning phase. That is, the model architecture for finetuning degenerates into a two-tower version that simply add up the outputs from the CTR model and PLM.
- **w/o Pretrain**. We remove the PA-MLM pretraining phase, while preserving the model architecture with soft prompt interface for downstream CTR prediction.
- **w/o Both**. We remove both the prompt interface and PA-MLM pretraining, which eliminates the interaction and alignment between collaborative and semantic knowledge during the training.

The results are reported in Table 5. When we remove either the prompt interface or PA-MLM pretraining, the performance degrades on three datasets for all backbone CTR models. This suggests that the explicit interaction and fine-grained alignment between collaborative and semantic knowledge can lead to better information extraction and fusion from both input modalities, and thus boost the CTR predictive performance.

## 4.5 Long-tail User/Item Analysis (RQ4)

The semantic information brought by PLM is especially valuable for scenarios with cold-start or long-tail users/items. Hence, in this section, we conduct in-depth analysis to further investigate the reasons for the performance improvement of ClickPrompt over the backbone CTR model from the long-tail user/item perspective.

We conduct the experiment on MovieLens-1M dataset with DCNv2 as the backbone CTR model and RoBERTa-base as the backbone PLM. We adopt the *finetuning-with-PLM* strategy. Specifically, we sort users/items based on their frequency of occurrence in the training set. The bottom 10% in terms of frequency are classified as long-tail low-frequency users/items, while the rest 90% are considered as non-long-tail ones. According to whether users and items are long-tail or not, we divide the entire testing set into four mutually exclusive subsets. We evaluate DCNv2 and ClickPrompt on each subset and report the results in Table 6, from which the following observations are obtained:

- Long-tail low-frequency users or items can lead to significant performance degradation for the traditional ID-based CTR model (*i.e.*, DCNv2), while ClickPrompt can consistently improves the predictive performance across all four subsets.

- In cases where the long-tail problem is more severe (*e.g.*, the subset where users and items are both long-tail), ClickPrompt can bring significantly larger improvements over the backbone CTR model. This confirms that ClickPrompt is effective in addressing cold-start or long-tail problems for recommendation, which mainly contribute to the final performance enhancement.

## 5 RELATED WORK

### 5.1 Traditional CTR Prediction

To estimate the user click probability, traditional CTR models usually convert the input data into ID features via one-hot encoding. The key idea is to capture the feature crossing patterns, which indicates the combination relationships of multiple features. While the implicit feature interactions are modeled by a deep neural network (DNN), the explicit feature interactions are captured by a specially designed learning function operator: (1) product operator, (2) convolutional operator, and (3) attention operator.

**Product operators** [12, 14, 19, 22, 48, 49] originate from classical shallow models such as FM [52] and POLY2 [4]. For example, DCN [55], xDeepFM [32], DCNv2 [56] are proposed to capture high-order feature interactions by applying product-based feature interactions at each layer explicitly. **Convolutional operators** [30, 37, 40] (*e.g.*, Convolutional Neural Networks (CNN) and Graph Convolutional Networks (GCN)) are also explored to capture the local and global views of feature patterns [37], and promote the interaction modeling through message propagation [30]. **Attention operators** [5, 29, 53, 60] suggest adopting the attention mechanism to allow feature fields or feature interactions to contribute differently to the final CTR prediction.

Although such an ID-based CTR modeling paradigm has achieved remarkable progress in the past decades, they generally suffer from the semantic information loss issue brought by one-hot encoding. This thereby leads to their disabilities to handle scenarios with cold-start users/items or low-frequency long-tail features.

### 5.2 PLM-based CTR Prediction

With the rapid development of pretrained language models (PLMs) in natural language processing (NLP) domains, researchers begin to explore the potential of PLMs for CTR prediction [33, 59]. Different from the ID-based one-hot encoding in traditional CTR prediction, the input data is converted into textual sentences through hard prompt templates as shown in Template A. According to the ground-truth label formulation and task genre, PLM-based CTR prediction can be roughly divided into two categories.

The first one [26, 45, 46] retain the ground-truth labels as binary codes {0, 1} similar to traditional settings, and model the CTR prediction task as a binary text classification problem. For instance, PTab [38] first further pretrains a BERT model [8] for the masked language modeling objective based on the textualized CTR data, and then finetune it for downstream CTR estimation with a randomly initialized prediction head.

The second category [36, 64] transforms the binary labels into a pair of key answer words (*e.g.*, Yes/No, Good/Bad), and thus models the CTR prediction as a sequence-to-sequence task. For example, P5 [10], as well as its variants [11, 17, 18], propose to tune T5 [51] as a unified recommendation model for various downstream tasks

in a textual generative manner. Other works [1, 39] also intends to incorporate decoder-only large language models (LLMs) to follow instructions and answer the user preference question appended after the textual input sentences.

Although the semantic information loss issue is well addressed, these PLM-based CTR models cannot capture the field-wise collaborative signals, leading to inferior CTR predictive performance. Moreover, the heavy inference overhead brought by the large model size makes it impractical for real-world industrial applications. Our proposed ClickPrompt could not only retain and fuse both the semantic and collaborative knowledge to achieve SOTA CTR predictive performance, but also deal with the inference inefficiency problem by providing a better semantic-aware parameter initialization for solely finetuning CTR model.

### 5.3 Prompt Tuning

Prompt tuning introduces a set of trainable continuous prompts to the pretrained language models for specific NLP tasks (*e.g.*, knowledge probing, text classification) [24, 41]. Generally, prompt tuning [13, 28] acts as a parameter-efficient finetuning (PEFT) solution, where we only update the soft prompts' parameters with supervision from downstream tasks, while keeping the entire parameters of the original PLM unchanged [25]. In this way, we can substantially reduce per-task storage and memory usage when finetuning PLMs on different downstream tasks. Moreover, some works [2, 42] propose to tune the soft prompts together with all of or part of the parameters of PLM. Notably, this setting no more belongs to the PEFT methods. It is very similar to the standard pretrain-finetune paradigm, but the addition of the learnable soft prompts can provide additional bootstrapping for model training [41]. Although this line of methods can significantly enhance the model capability, it requires heavy computational and storage resources and may overfit on small datasets.

In this paper, ClickPrompt adopts the basic idea of prompt tuning [41, 47] to connect the CTR model and PLM with the layerwise trainable soft prompts. In general, the structure of the CTR model is specially designed to capture essential feature interaction patterns, and is therefore a naturally strong soft prompt generator to adapt PLMs to the downstream CTR prediction task.

## 6 CONCLUSION

In this paper, we propose a novel model-agnostic framework (*i.e.*, ClickPrompt), where CTR models serve as the soft prompt generators for PLMs. A pretrain-finetune scheme is designed to enable explicit interaction and alignment between the collaborative knowledge from one-hot ID modality and the semantic knowledge from textual modality, which significantly improves the CTR predictive performance. Furthermore, we provide another lightweight finetuning strategy to solely train the CTR model for downstream tasks without PLMs, thus properly tackling the inference inefficiency issue. Extensive experiments on four real-world datasets validate the superior predictive performance and model compatibility of ClickPrompt compared with baseline models. As for future works, a promising direction is to further improve pretraining efficiency. Moreover, we will explore the application of ClickPrompt on other recommendation tasks (*e.g.*, learning to rank).

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

## A    DATA PREPROCESSING

We conduct experiments on four real-world datasets from different recommendation scenarios. The information about data preprocessing is given below:

- **Movielens-1M** is a movie recommendation dataset from Movielens website with ratings ranging from 1 to 5. We binarize the ratings with a threshold of 4, while removing neutral samples with ratings equal to 3 [31, 53].
- **BookCrossing** is a book recommendation dataset from BookCrossing website with ratings ranging from 0 to 10. We convert the ratings into binary labels with a threshold of 5.
- **Amazon-Toys** is a e-commercial dataset of toys category from Amazon with ratings ranging from 1 to 5. We binarize the ratings with a threshold of 4. We apply 5-core filtering to ensure each user or item has at least five interaction records [10, 31].
- **GoodReads** is a book recommendation dataset from GoodReads website with ratings ranging from 1 to 5. We transform the ratings into binary labels with a threshold of 4. Similar to Amazon-Toys, we apply 10-core filtering to ensure each user or item has at least ten interaction records.

## B    BASELINE IMPLEMENTATION

In this section, we give the hyperparameter configuration for each baseline model from two different categories: (1) traditional CTR models, and (2) PLM-based CTR models.

### B.1    Traditional CTR Models

We train each traditional CTR model from scratch based on click signals without pretraining. Similar to the finetuning stage of ClickPrompt, we adopt AdamW [44] as the optimizer. The batch size is set to 256 for Movielens-1M, 256 for BookCrossing, 1024 for AZ-Toys, and 4096 for GoodReads. The learning rate is set to $1 \times 10^{-3}$. We set the embedding size to 32 for MovieLens-1M and BookCrossing, and 16 for Amazon-Toys and GoodReads. The dropout rate is selected from $\{0.0, 0.1, 0.2\}$. We utilize one linear layer after the feature interaction layer to make the final CTR prediction. Unless stated otherwise, we adopt ReLU as the activation function. The model-specific hyperparameter settings for base models are as follows:

- **DNN**. We select the size of DNN layer from $\{128, 256, 512\}$, and the number of DNN layers from $\{3, 4, 5, 6\}$.
- **DeepFM** [12]. We select the size of DNN layer from $\{128, 256, 512\}$, and the number of DNN layers from $\{3, 4, 5, 6\}$.
- **xDeepFM** [32]. We choose the number of CIN layers from $\{2, 3, 4, 5\}$, and the number of units per CIN layer is set to 25. We select the size of DNN layer from $\{128, 256, 512\}$, and the number of DNN layers from $\{3, 4, 5, 6\}$.
- **IPNN** [48]. We select the size of DNN layer from $\{128, 256, 512\}$, and the number of DNN layers from $\{3, 4, 5, 6\}$.
- **DCN** [55]. We select the size of DNN layer from $\{128, 256, 512\}$, and the number of DNN layers from $\{3, 4, 5, 6\}$. We force the CrossNet module to have the same number of layer as the DNN network.
- **AutoInt** [53]. We select the number of attention layers from $\{3, 4, 5, 6\}$. The number of attention heads per layer and the attention size are set to 1 and 32, respectively.
- **FiGNN** [30]. We select the number of layers from $\{3, 4, 5, 6\}$, and apply residual connection for the graph layers.
- **FGCNN**. We maintain 4 tanh-activated convolutional layers with a kernel size of 7 and pooling size of 2 for each layer. The number of channels for each layer is set to 6, 8, 10, 12, respectively. The numbers of channels for recombination layers are all set to 3.
- **DCNv2** [56]. We select the size of DNN layer from $\{128, 256, 512\}$, and the number of DNN layers from $\{3, 4, 5, 6\}$. We force the CrossNet module to have the same number of layer as the DNN network.

### B.2    PLM-based CTR Models

This line of methods generally incorporate the pretrained language models for CTR prediction. We keep the structure of PLMs unchanged, and describe the training setting for each model as follows. Note that we utilize AdamW [44] as the optimizer for all of the PLM-based baseline models.

- **CTR-BERT** [46]. We maintain a two-tower model structure based on the BERT [8] model to encode the user and item information, respectively. We set the total number of tuning epochs

to 10 with batch size of 1024. The learning rate is set to $5 \times 10^{-5}$ with linear decay. The warmup ratio is set to 0.05.

- **P5** [10] is a unified sequence-to-sequence framework with T5 [51] as the backbone pretrained language model for multiple recommendation tasks. In this paper, we leverage P5 for a single task only (*i.e.*, CTR prediction). The total number of epochs is set to 10 with batch size of 32. The learning rate is selected from $\{5 \times 10^{-4}, 1 \times 10^{-3}\}$ with linear decay. The warmup ratio is 0.05. Following P5's official implementation, we also perform gradient clip with threshold equal to 1.0.
- **PTab** [38] adopts the common pretrain-finetune scheme based on the BERT [8] model. PTab first further pretrains the BERT model with the classical masked language modeling objective based on the textualized CTR data, and then finetunes BERT for downstream CTR prediction as a text classification problem. Following the original paper, we pretrain BERT for 10 epochs

with batch size equal to 1024. The learning rate for pretraining is set to $5 \times 10^{-5}$ with linear decay. The warmup ratio is 0.05. As for finetuning, the total number of tuning epoch is set to 10 with batch size of 1024. The learning rate for finetuning is initialized at $5 \times 10^{-5}$ with linear decay. The warmup ratio is 0.01.

- **CTRL** [27] designs a contrastive pretraining framework to implicitly align the collaborative knowledge from CTR models and semantic knowledge from PLMs. Following the original paper, we choose AutoInt as the backbone CTR model and TinyBERT [21] as the backbone PLM. We first perform contrastive pretraining for 20 epochs, and then finetune AutoInt for downstream CTR prediction tasks. The model structure of AutoInt is set as the best configuration based on the grid-search result stated in Appendix B.1. Other training configurations are the same as reported in CTRL's original paper [27].

