# OpenReview forum: "ClickPrompt: CTR Models are Strong Prompt Generators for Adapting Language Models to CTR Prediction"
_ACM.org/TheWebConf/2024/Conference — TheWebConf24 Oral_

### Official Review · Reviewer_XRfr · 2023-11-04

**Novelty:** 5
**Technical Quality:** 5

**Review:**

The paper discusses the importance of click-through rate (CTR) prediction in various internet applications. It highlights the limitations of traditional CTR models that suffer from semantic information loss and explores the use of pretrained language models (PLMs) but points out their inefficiency in capturing collaborative information and their large model size.

The main contribution of the paper is the introduction of a novel framework called "ClickPrompt." This framework combines the strengths of CTR models and PLMs to accurately estimate CTR while addressing the inference inefficiency issue. ClickPrompt uses CTR models to generate interaction-aware soft prompts for PLMs, enabling the alignment and interaction of collaborative and semantic knowledge from different types of features. This approach allows for improved CTR estimation and provides the flexibility to either fine-tune the CTR model with PLM for enhanced performance or fine-tune the CTR model independently for improved inference efficiency.

Experiments conducted on real-world datasets validate the effectiveness of ClickPrompt when compared to existing baseline methods. The paper provides source code for further exploration and implementation of the proposed framework.

Reasons To Accept:

1. The writing of this paper is clear and easy to understand.
2. The method is simple but effective, achieving competitive or superior performance, especially compared to previous baselines.
3. Choice of datasets and experiments is thorough.
4. The idea of prompt-augmented masked language modeling is interesting, and the ablation study results seem to suggest that the PA-MLM pretraining phase is important.

Reasons To Reject:

1. In my opinion, the RQ2 is not fully answered. Besides the RoBERTa-base, the authors only experiment with encoder-only models TinyBERT and RoBERTa-large. However,  the model compatibility should not only consider the model size, but also the model architecture such as encoder-decoder models (e.g., BART and T5) and decoder-only models (e.g., BLOOM and LLaMA).
2. The idea of adapting language models to CTR prediction is interesting. However, I am not quite convinced that PLM plays an important role in the CTR process or the soft prompts generated by CTR models are “good prompts for PLM”. This should be explicitly discussed in the paper.
3. Although ClickPrompt has been proven effective in experiments, the improvement compared to the baseline is actually very small, mostly less than 0.5%.
4. Finetune CTR models without PLMs is for the sake of inference efficiency. However, there are no experiments or discussions on efficiency and time in the paper.

Typo:
Line 557: for each baseline models -> for each baseline model

**Questions:**

1. Is it possible to apply ClickPrompt to other large language models? If possible, why did this paper only use the encoder-only model? How can we better utilize the knowledge of PLMs in the era of LLMs?

2. It seems like that the collaborative knowledge is much more effective than semantic knowledge in CTR prediction (Please correct me if I am wrong). What is the underlying reason behind this phenomenon? Can't PLM make good use of this knowledge?

3. I am curious about the performance of ChatGPT in predicting CTR. The authors are encouraged to consider adding discussion on this aspect in the experiment.

**Ethics Review Description:**

No ethical issues.

**Reviewer Confidence:**

3: The reviewer is confident but not certain that the evaluation is correct

**Scope:**

4: The work is relevant to the Web and to the track, and is of broad interest to the community

---

### Official Review · Reviewer_Qk5s · 2023-11-16

**Novelty:** 4
**Technical Quality:** 4

**Review:**

**Summary:**
The paper, "ClickPrompt: CTR Models are Strong Prompt Generators for Adapting Language Models to CTR Prediction," proposes a novel framework that integrates click-through rate (CTR) models with pretrained language models (PLMs) to enhance CTR prediction. This integration addresses the semantic information loss in traditional CTR models and the inefficiency of PLMs in capturing collaborative knowledge. The framework, called ClickPrompt, uses CTR models to generate soft prompts for PLMs. These prompts facilitate the alignment of semantic and collaborative knowledge through a prompt-augmented masked language modeling (PA-MLM) pretraining task.

**Strengths:**
1. **Innovative Approach:** ClickPrompt's approach to combining CTR models with PLMs for better CTR prediction is innovative. It creatively addresses the limitations of both approaches when used independently, proposing a symbiotic relationship where each compensates for the other's weaknesses.
2. **Addressing Semantic Loss and Inefficiency:** The paper effectively identifies and tackles two critical issues in CTR prediction: the semantic information loss in traditional CTR models and the inefficiency of PLMs in handling collaborative knowledge. This dual focus enhances both the accuracy and practicality of the proposed solution.

**Weaknesses:**
1. **Handling Complex Feature Interactions:** While ClickPrompt addresses semantic and collaborative knowledge, how it handles complex feature interactions within these domains could be further elaborated. Future work might focus on explicating these interactions, especially in more nuanced or complex data environments.
2. **Generalizability and Scalability:** Given the integration of two complex models, questions about scalability and computational efficiency arise. The paper could benefit from a more detailed discussion on how ClickPrompt scales with larger datasets and its computational requirements.
3. **Generalizability Across Diverse Domains:** While the paper tests the framework on four datasets, further exploration of its applicability across more diverse domains and different types of CTR prediction scenarios would strengthen its generalizability claims.
4. **Long-term Adaptability:** In rapidly evolving fields like CTR prediction, the long-term adaptability of models is crucial. Future iterations of the framework could focus on how it adapts to new data patterns and evolving user behaviors over time.

**Questions:**

1. How does ClickPrompt perform in real-time, resource-constrained environments?
2. Can the approach be effectively generalized to different types of recommendation systems beyond those tested?
3. What are the specific trade-offs when choosing between the two finetuning strategies of ClickPrompt?

**Reviewer Confidence:**

4: The reviewer is certain that the evaluation is correct and very familiar with the relevant literature

**Scope:**

4: The work is relevant to the Web and to the track, and is of broad interest to the community

---

### Official Review · Reviewer_x81e · 2023-11-22

**Novelty:** 5
**Technical Quality:** 5

**Review:**

The authors propose a novel Prompt-based method that facilitates implicit interaction between task-specific models and pre-trained models through collaborative knowledge and sentiment knowledge. Two fine-tuning methods were devised: one aimed at enhancing CTR performance without altering the CTR model structure, and the other aimed at improving CTR performance without incurring additional inference costs.
Initially, this approach utilizes a Prompt generation layer within the CTR model to produce trainable soft prompt vectors. Subsequently, an enhanced Masked Language Modeling (PA-MLM) task is devised for pre-training. Leveraging BERT's token masking strategy, text features are masked, and PLM (Roberta) is employed to reconstruct the corrupted text features based on textual context and soft prompts. The soft prompts serve as a bridge, facilitating the forward propagation of ID-based collaborative knowledge to PLM (using soft prompt vectors as prefix state inputs in each layer of PLM), while the text-based semantic knowledge in PLM is propagated back to the CTR model through backward propagation. Semantic and collaborative knowledge pertaining to both ID and text align and interact via the prompt interface, enabling implicit interaction between collaborative and semantic knowledge. This interaction enhances the CTR performance without altering the CTR model structure or incurring additional inference costs.
Following pre-training, the authors propose two distinct CTR fine-tuning strategies:
Fine-tuning using PLM: Adjusting both the CTR model and PLM as a unified entity by connecting them through the prompt generation layer, resulting in superior performance.
Fine-tuning without PLM: Addressing inference efficiency concerns by solely fine-tuning the CTR model without PLM. The PA-MLM pre-training provides semantic-aware parameter initialization for downstream CTR fine-tuning, promoting ultimate performance without altering the CTR model structure or incurring additional inference costs.

**Questions:**

The MLP module in line 219 is used without a specific explanation for the abbreviation.
As you mentioned in Prompt Generation, the prompt vector 𝑝𝑖 is from MLP. To complete the cloze task over masked tokens, how does PLM extract and incorporate the corresponding “right answer” embedded in the soft prompts?
The difference between qi in Equation 10 and qi in Equation 5 hasn't been addressed.
Is the time cost considered when fine-tuning using PLM?

**Reviewer Confidence:**

1: The reviewer's evaluation is an educated guess

**Scope:**

4: The work is relevant to the Web and to the track, and is of broad interest to the community

---

### Official Review · Reviewer_DoUU · 2023-11-26

**Novelty:** 3
**Technical Quality:** 4

**Review:**

The authors present a training strategy to harness the strength of Pretrained Language Models to improve CTR model's performance. The idea presented seems to be one way to combine, but it is not clear if it is computationally efficient to integrate LLMs into CTR models. Would've liked to see the size of the PLMs employed in the experimental setup to understand the computational cost of training such models.

**Questions:**

Are the comparative models used in evaluation contain similar number of parameters to train? The reason I was curious about this is because if the base models are small, they wouldn't match the performance of the ClickPrompt model since it is using the PLM.

**Reviewer Confidence:**

2: The reviewer is willing to defend the evaluation, but it is likely that the reviewer did not understand parts of the paper

**Scope:**

3: The work is somewhat relevant to the Web and to the track, and is of narrow interest to a sub-community

---

### Official Review · Reviewer_kXZj · 2023-11-27

**Novelty:** 5
**Technical Quality:** 5

**Review:**

This article addresses the issue of improving the accuracy of click-through rate (CTR) prediction tasks. Traditional CTR models suffer from the problem of semantic information loss, while the method of using pre-trained models with hard prompt templates fails to capture collaborative information. In order to simultaneously capture semantic knowledge and collaborative knowledge, achieve more accurate CTR estimation, and solve the problem of low inference efficiency in Pre-trained Language Models (PLMs), a new framework called ClickPrompt is proposed, along with its specific pre-training tasks. This framework is compatible with various CTR and PLM models, and it demonstrates state-of-the-art results on four real-world public datasets.

**Questions:**

1. You claim to have proposed modeling semantic knowledge and collaborative knowledge, but CTRL[1] is an earlier work this year that also mentions related content. Although the results of this article surpass CTRL in the experiments, no explanation of the differences between the two is found in this article. I hope the authors can provide some clarification on this.
2. The technical contribution is limited, as the PA-MLM pre-training task does not differ significantly from the MLM pre-training task and is a relatively intuitive approach.

[1]	Xiangyang Li, Bo Chen, Lu Hou, and Ruiming Tang. 2023. CTRL: Connect
Tabular and Language Model for CTR Prediction. arXiv preprint arXiv:2306.02841
(2023).

**Reviewer Confidence:**

3: The reviewer is confident but not certain that the evaluation is correct

**Scope:**

3: The work is somewhat relevant to the Web and to the track, and is of narrow interest to a sub-community

---

### Decision · Program_Chairs · 2024-01-22

**Decision:**

Accept (Oral)

**Comment:**

This paper presents ClickPrompt, which incorporates click-through-rate (CTR) models to generate soft prompts for pre-trained language models (PLMs). These prompts facilitate the alignment of semantic and collaborative knowledge through a prompt-augmented masked language modeling (PA-MLM) pre-training task. The idea of integrating collaborative information and semantic information for CTR prediction is novel, and the proposed solution is elegant and effective. The paper also explored two distinct CTR fine-tuning strategies, i.e., Fine-tuning with PLM and Fine-tuning without PLM. Experiments are conducted on four public datasets from different domains and varying from different scales, and also compared with both CTR models (AutoInt, DCN, etc.) and language models (P5, BERT, etc.) as baselines, which are comprehensive. The paper received comprehensive discussions between reviewers and authors. During the discussions, authors made sufficient clarification and also provided experimental results requested by reviewers to better illustrate the work, which addressed reviewers' questions in satisfactory ways.